# Basic Conditions for Support of Young Carers in School: A Secondary Analysis of the Perspectives of Young Carers, Parents, Teachers, and Counselors

**DOI:** 10.3390/healthcare12111143

**Published:** 2024-06-04

**Authors:** Steffen Kaiser, Steffen Siegemund-Johannsen, Gisela C. Schulze, Anna-Maria Spittel

**Affiliations:** 1Department of Education of People with Intellectual and Developmental Disabilities, Institute for Special Education, Europa-Universität Flensburg, 24943 Flensburg, Germany; steffen.kaiser@uni-flensburg.de (S.K.); steffen.siegemund-johannsen@uni-flensburg.de (S.S.-J.); 2Department of Special Needs Education and Rehabilitation, Special Needs Education, Rehabilitation and Health Care, Carl von Ossietzky Universität Oldenburg, 26129 Oldenburg, Germany; gisela.c.schulze@uni-oldenburg.de

**Keywords:** young carers, young adolescent carers, school, education, support

## Abstract

Young carers face a variety of challenges at school. While schools can be vital places of support, the assistance they receive at school often seems selective and fails to consider the unique life situations of individual students. This paper examines the perspective of multiple actors in the student’s school environment and explores how schools can develop comprehensive, sustainable support systems for young carers—systems that consider and involve as many actors as possible in the student’s school environment. In a secondary analysis of two interview studies, we analyzed how young carers as well as their parents, teachers, and school counsellors perceived the school support the carers received. We then developed an integrated model that incorporates these differing perspectives. The model offers an approach for implementing low-threshold support for young carers within existing school structures in relation to their family situation and outlines conditions that can support both recognized and “invisible” young carers, as well as other students.

## 1. Introduction: Young Carers in School and Potential Support Resources

Illness and impairment of a family member affect the entire family, including its youngest members. Children and young people are often involved in providing care and support services, thereby functioning as what professionals refer to as “young carers”. Becker [1] defines young carers as “children and young persons under 18 who provide or intend to provide care, assistance or support to another family member. They carry out significant or substantial caring tasks, and often regularly assume a level of responsibility usually associated with an adult. The person receiving care is often a parent but can be a sibling, grandparent or another relative who is disabled, has some chronic illness, mental health problem or other condition connected with a need for care, support or supervision” (p. 378). In this role, children and young people take on a variety of tasks, including physical, emotional, and medical or therapeutic support for the sick person; household activities; tasks outside the household; support and care for healthy family members; and responsibility for themselves [2]. The range of activities can vary from occasional assistance to being solely responsible for providing round-the-clock care [3]. Young carers are a global reality, and the young carer role itself is comparable across countries [4]. However, awareness, research, support structures, and policy responses differ between countries, with the UK currently categorized as advanced, Sweden as intermediate, Germany as preliminary, and the United States as emergent in these areas [5].

Young carers view school as important for their lives, appreciating its potential to open up opportunities to learn and experience new things [6,7,8,9]. Research indicates that young carers assign various meanings to school and may see it as a place of identification [6,10,11], a source of support [9,11,12], a venue for connecting with others in similar situations [6,9], a “break from responsibilities at home” [13] (p. 324), or a “safe haven” [14] (p. 532).

However, some young carers face challenges in school due to their family situations. These can manifest in the form of patchy school attendance, lack of concentration, poor academic performance, problematic behavior, emotional issues, social isolation, and limited social interactions [8,15,16]. 

As schools often serve as the initial point of recognition for young carers’ situations, they are ideally positioned to initiate support for these students [17]. Nevertheless, the level and type of support young carers receive in schools can vary greatly. While some young carers receive no support, others benefit from diverse forms of assistance from various individuals or groups. This can include direct help from classroom teachers, aid from counseling services or group counseling activities, or support and feedback from other young carers during informal encounters with them [6,9,11,12,18]. Schools can also be a valuable source of information on issues like illness, disability, or mental health and can facilitate external support services—not only for young carers, but also for other children [6,11,12,18]. 

Support for young carers tends to differ from country to country [5,17], which also affects in-school support structures. One clear assumption is that differing school systems lead to entirely distinct structures. However, studies also point to broader discrepancies between countries when it comes to awareness about young carers and policy responses to them, which have consequential effects on support structures [5]. A lack of awareness often leads professionals to overlook the background situation of young carers and thus fail to identify them as such. These professionals may recognize certain signs but not associate them with the phenomenon of young carers [17,19,20,21]. Furthermore, schools themselves tend to evaluate student issues as freestanding incidences or problems, rather than assessing them within a broader context. This is particularly evident in the case of young carers, whose situations are often invisible to others in school settings [18]. Finally, different stakeholder groups—parents, teachers, counsellors, and the students themselves—may have conflicting perceptions of young carers in terms of the situations and school-based support they receive. In this vein, Kaiser [18] has analyzed how parents, teachers, and young carers themselves perceive the situation of young carers in schools, noting that conflicting views among these actors can lead to support that is insufficient, inadequate, and short-term.

When it comes to optimizing support strategies, research shows that strategies solely directed at young carers are often insufficient in the long term [17,22]. Indeed, comprehensive and sustainable support requires the involvement of all actors in a young carer’s environment [18,23]. Thus, the prevailing paradigm in the international literature [24] advocates for a “whole family approach” [25], highlighting the positive long-term impact on school situations of an approach that considers the entire family, rather than the carer in isolation [22]. The aim is to create a multiprofessional network that engages students, their families, and their personal surroundings [23]. As research indicates, these supportive frameworks must be recognized by everyone within an educational institution to ensure their sustainability [17]. This raises the question of how to design school-based support structures that will be accepted by all actors in school. 

This issue is still understudied. Previous studies on young carers in educational settings mainly focused on individual stakeholder groups [8,16,26,27,28,29], and few have compared the views of different individuals and groups the who intersect with young carers at school [6,18,21,30]. Furthermore, no studies have examined the perspectives of school counselors, despite the key professional role they play within school support structures. Research on young carers in schools has mainly focused on how caregiving affects their school performance, especially with reference to school problems. Support is only discussed as one factor among many [8,16,18,21,28,30].

In light of these considerations, this article aims to analyze the perceived school support from four key stakeholders in schools: young carers, parents, teachers, and school counselors. After initially analyzing each perspective separately, the resultant findings are integrated and the results of this novel approach interpreted to identify the basic conditions of support present in schools. Analysis of the four stakeholder groups allows us to develop and discuss a framework model that can be used specifically to support young carers within their school environments.

## 2. Materials and Methods

The research methodology employed in this study is qualitative secondary analysis. This approach re-examines existing data originally collected for other research purposes [31]. In the present study, these data (collected by two of the authors of the two primary research studies) are examined as a complement to those projects [31]. A supplementary method of analysis was used [31].

### 2.1. Description of the Re-Used Data

The data used in this study are derived from two different research projects conducted in Germany.

**Research Project 1** involved interviews with young carers (*n* = 9), their parents (*n* = 8), and teachers (*n* = 4). Plans to conduct an additional interview for each young carer (involving the young carer, a parent, and one of the teachers) were abandoned due to lack of feedback and young carers’ wish not to interview teachers, among other constraints. The interviews took place between November 2015 and September 2016 using a semi-structured format [32]. Conducted in German and in person by the first author of this article, they included individual interviews and one double interview with two teachers (a class teacher and a special needs teacher). Participants were interviewed separately at locations chosen by them (either at home or, in the case of some teachers, at school) and recorded using a digital audio recorder. This was subsequently transcribed in full by the first author in standard orthography. Non-verbal communication was only integrated if it appeared important for the content of the interviews. The young carers we studied in Project 1 were aged between 13 and 16 at the time of the interviews (grades 7 to 11) and lived in different family constellations, where they looked after siblings, parents, grandparents, or other relatives. They adopted a level of care which could be defined as at least moderate (the MACA-YC18 [33] score provided by the young people ranged between 12 and 15). 

Research Project 1 was Germany’s first study to focus on young carers and education, for which it used a field theory approach. Its underlying aim was to gain exploratory insight into the school situation of young carers. Its primary aim was to identify the factors that influence the educational settings of young carers. To that end, the analysis was segmented into sub-sections that covered the perspectives of each of the three stakeholder groups. The secondary aim was to ascertain the nature of the support provided in schools. All three interview guides covered four main subjects: young carers’ family and school situations; communication; and the perceived support they received. The respondents were prompted to recount their perceptions on each topic by means of open-ended questions. The guide also incorporated sub-questions for exploring relevant themes, as needed [18]. Through the analysis, comprehensive explanatory accounts detailing the school-related experiences and situations of young carers were developed. Various factors influencing these situations were identified. The results indicate several potential approaches to supporting young carers in school within the contexts of family, school, and society [18]. 

In this supplementary analysis, we re-examine the interviews conducted with the three stakeholder groups. Here, we focus solely on the school and the success or failure of the support provided.

**Research Project 2** focused the perspective of school counselors on the issues surrounding young carers and their role as support resources for students who fall into this group. The school counseling system, which underwent analysis, supports students, parents, teachers, and schools as a whole with a variety of problematic situations at school. This can involve not only individual students and their parents, but also entire classes or schools. The participants, referred to here as “counselors”, came from different professional backgrounds. They included special needs teachers, psychologists, social education workers, and regular teachers who received counseling training. The overall aim of the project was to determine how the issue of young carers fits into the counseling system. For this, we used a mixed-methods design, comprised of a quantitative section (a survey) plus qualitative interviews. The present secondary analysis draws on a part of the project that took place between February 2022 to August 2022, when we conducted individual semi-structured interviews [32] in German with school counselors (*n* = 8). The purpose of the interviews was to investigate the counselors’ potential role in identifying young carers during counselling sessions, intervening in school-related problems, and preventing such issues from occurring. The focus was on the professional setting in which counselors worked and their role as a support resource for young carers [34]. With the exception of one interview that took place by phone at the participant’s request, all of the interviews were conducted online by the last author of this article using the web conferencing software BigBlueButton (https://meeting.uol.de/b). In all of the online interviews, only the audio track was recorded on the computer via background software. The interviews were then transcribed in full by the author in standard orthography; non-verbal communication was only integrated if it appeared important for the content of the interviews. 

In the current analysis, we re-examine these data to identify factors mentioned by the counselors in relation to the support options for teachers in schools. 

In this article, we re-examine the data of the two studies (interviews with young carers (*n* = 9), their parents (*n* = 8), their teachers (*n* = 4), and counselors (*n* = 8) in relation to the perceived support options in school. The two research projects were selected for secondary analysis because they represent two of the few qualitative studies in Germany that focus on young carers in school contexts, and because both studies share the same theoretical basis. The purpose of this secondary analysis is to analyze the perceived school support from four key stakeholders in schools and, more broadly, to address the question: What are the basic conditions for successful support structures for young carers in schools?

Both studies draw on Lewin’s field theory, which seeks to explain human behavior in the context of the environment [35]. Participants are seen as integral parts of their environment and viewed within the context of their interactions with it. The analysis is based on the subjective perspective and the construction of a person’s life space [35]. A key assumption of this theory is that every aspect of an individual’s reality shapes their behavior. Research Project 2 also utilizes Bronfenbrenner’s ecological theory [36], which is grounded in Lewin’s work, to present and analyze its findings at various levels.

This field-theoretical approach aligns with the core principles of young carers research, recognizing the target group and their family as experts and addressing their perspectives in both research and practice [25]. Accordingly, we prioritize the individual perceptions of each participant to ensure a comprehensive approach. 

**Ethics:** Ethical approval for both studies was obtained from the research ethics committee at the Carl von Ossietzky Universität Oldenburg. All participants were verbally briefed on the study’s objectives and content prior to their participation. They also received individual information letters that were tailored and linguistically adapted for each target group. After receiving answers to all their questions, all participants gave written consent to participate in the study by signing a consent form. Parents also signed a declaration regarding the involvement of their children in the study. The mother of one young carer was notified in writing and confirmed their child’s participation in written form. The study adhered to ethical guidelines outlined by the German Psychological Society (DGP). No incentives for participation were provided.

### 2.2. Analysis of Data

To achieve the objective of the study, a focused interview analysis [37] was conducted using all 29 interviews.

Initially, all prepared interviews were compiled into a joint MAXQDA project. This was followed by a preliminary exploration of all the data by two of the authors (Step 1) [37]. The initial coding phase (basic coding) aimed to identify factors linked to effective or ineffective support for young carers in schools, classified into four perspectives (Step 2 and 3) [37]. A category system was then inductively developed for each perspective. During the coding process, it became clear that many statements about school support were associated with one of two contexts: the classroom or the school context. Additional focal points emerged during the interviews and were recorded as main categories based on the feedback of individual participants (see Table 1). Notably, “bullying and how it is handled” was not categorized as a school subcategory of school context but was included as a separate main category due to its significance in the interviews. During coding, numerous sub-categories were inductively created for these main categories. The interviews were then finely coded using this category system (Step 4) [37]. The results presented reflect the perspectives of each group on issues within the main thematic categories. Due to space limitations, we have reformulated subcategories as themes in the text. These are italicized in the results section (Section 3.1). 

The results of this coding process are detailed in Section 3.1 (Step 6) [37]. The findings are then interpreted and consolidated into a model (Step 5) [37].

## 3. Results

The following section presents our results, organizing them by participant group (young carers, parents, teachers, counselors). Structured around main categories, the presentation details the participants’ statements about support in schools and their perceptions of its sufficiency or insufficiency. These findings are then interpreted from all four perspectives. From this we identify the basic conditions for supporting young carers at school, which are then synthesized into a model.

The following interview excerpts were translated into English by the authors and corrected by an English-speaking editor. The intention is to retain the language style as much as possible.

### 3.1. Presentation of Results

#### 3.1.1. Perspective of the Young Carers

**Classroom context:** The young carers recognize a *teachers’ consideration* when they know about the family situation. Their statements reveal a range of valuation of such consideration: direct interventions by teachers, such as adjusting the homework amount, lead to alleviations in school and are positively appraised by the young carers. Some teachers would notice when they are particularly stressed at school and refer them to a quieter learning environment. However, some young carers dislike being treated differently by their teachers, especially when they get the feeling of compassion: 


*So, I kind of had the feeling before, they treated me a little bit differently. So like: “oh you poor thing”—something like that. And now just nothing. (…) So I think it’s rather better that way (YC2).*


Of all their teachers, the carer’s classroom teacher is seen as the primary support figure. An emerging pattern is the students’ *confidence* in classroom teachers, which is not extended to other teachers. As this YC3 said:


*I really only talk about it with Mrs. Schultz. No one else, because she’s my classroom teacher.*


The mere *possibility to talk to teachers about the family* is perceived as positive. Furthermore, teachers’ specific enquiries about a carer’s family situation are seen as supportive. Communication is not solely initiated by young carers, nor do young carers share everything with their classroom teachers. For example, YC3 is selective when it comes to discussing her mother’s mental illness:


*I want to keep this private now and not really bring up issues anymore, not even with Mrs. Schultz, or with the students, or the teachers in general, I don’t really intend to (YC3).*


**Bullying and how it is dealt with:** Bullying is a recurrent theme in the interviews with young carers. Many participants explicitly link bullying to illness or disability, whether their own, a family member’s, or that of others. Participants point to a lack of understanding in the classroom as an important driving force for this behavior:


*The other students laugh at them [note: fellow students with disabilities] because one of them has a paralysis on one side. And there is one student […] who looks different from the others, and they laugh at them and that’s just not right. But that’s just the problem, that they are not educated at all (YC2).*


As a result, this participant gave a talk about their siblings’ disabilities to educate their peers, noting a change in classmates’ behavior afterwards. However, YC2 notes the difficulty of the process leading up to this change and highlights the significance of trust in class:


*Yes, I think this education is actually very important. Well, my mother also has a disease and I think you should just, I don’t know, make a day where everyone just… The problem is, you need this trust in the class and very few people actually have that. In our class you can’t say everything with some people. And I think that’s rather the biggest problem, because that’s why a lot of people are afraid to say anything (YC2).*


Several strategies for *intervening in cases of bullying and discrimination* are mentioned. Some young carers indicate the availability of a suggestion box when communication in class is not feasible. Students can use the box to anonymously share their own burdens or other issues, which are then addressed in a class council. Overall, teachers’ responses to bullying are perceived as inadequate.


*Because she [note: the classroom teacher] has now also said… I don’t think she understands that either, because she said, “Everybody who was involved in bullying YC3 has to write a letter of apology to me now.” I got 13 letters of apology! Yes. There’s a lot in there, but, sure, they’re not sorry at all, I can tell. They just don’t want to get shit on, yeah (YC3).*


Young carers also highlight the importance of language in the context of bullying and disability:


*[…] because there are these disabled children, or we also have special children and when they are teased or so (…). [Interviewer: Who are the special children?] YC5: Well, just the disabled children, but I think that one should not say: You are disabled, but just that one says special (YC5).*


Again, *peer education* is seen as the basis for change:


*This “disabled” is always such a word, you can’t really say it. And that’s what I told them. And since then, they have also been very reserved about the word and have accepted it well (YC2).*


#### 3.1.2. Perspective of the Parents

**Classroom context:** Most parents acknowledge that their children’s home care responsibilities, especially during acute periods of illness, create burdens on them at school and in the classroom. While some perceive negative effects, others notice no impact, instead highlighting their children’s positive attributes, like self-confidence and initiative. Two mothers mention their daughters’ classroom talks about their siblings’ disabilities, both *to raise awareness among their peers* and to process the topic for themselves. As both of these mothers note, the talks also educated the teachers:


*[…] that her classroom teacher stood in front of her crying during her talk, saying: “Wow, I didn’t know that.”(P7)*


But not all young carers are willing to approach the topic openly. A range becomes clear: there are children who deal with it frankly and children who feel ashamed of their relatives. As P2 explains when talking about children of disabled siblings:


*Some, they think it’s totally embarrassing to drive around with a kid in a wheelchair. Or when he or she drools or something like that. Then they avoid bringing schoolmates home and others just the opposite. And you somehow can’t influence it like that, actually not at all (P2).*


All parents express the desire for teachers to *recognize, acknowledge, and encourage the unique abilities of each child* in the classroom. They note that effective support for their child depends on teachers’ *awareness of the family situation.* Many parents find it challenging to communicate on the subject. They recognize their personal responsibility in the matter, but acknowledge that other matters take priority, particularly at the onset of an illness, when a family coping process ensues. Thus, teachers are not always immediately informed about the situation at home. All the parents we interviewed had discussed their family situation with teachers on various occasions, such as during illness flare-ups, new therapies, or the death of a family member. The parents stress that they do not want their children to receive preferential treatment, but rather more consideration from teachers:


*I don’t want you [note: the teacher] to favor them [note: his kids] at all, just being aware. And know that if a YC1 ever freaks out, that there are other reasons. And be a little gentler with them and see it from a little different perspective (P1).*


As a sign of their children’s stress, parents see an atypical change in the child’s behavior in class and school:


*However, the aggressive behavior that she had for a while, just briefly, that was probably just this cry for attention, for help. That was probably a cry for help, I think. That’s how you could interpret it. Because … And there are a lot of kids who are like that, right? Who have this behavior in school. And I’m sure there’s always going to be a reason why a kid freaks out like that or gets bad at school, closes themselves off from everybody else and becomes a loner or something (P3).*


One mother interprets her daughter’s behavior as a way of expressing at school “what her life actually is like” (P7). Parents expect that *teachers respond to the changed behavior appropriately*. However, they note that many teachers are incapable of doing so.

**School context:** The interviewed parents mention the existing and wished-for support mechanisms in school. One important factor noted by these interviewees is the *need for schools to provide information* that is accessible to both students and parents, perhaps through an anonymous noticeboard. They also suggest that schools should designate (or communicate to students and parents, if one has already been designated) a *contact person for young carers.* Parents see teachers or social workers in school as contact persons for issues concerning young carers. 

Schools need to become places of encounter. Some parents see networking with other parents as particularly important:


*That’s why I also say that working with other parents at school is the be-all and end-all. That you at least find each other and exchange ideas, because at the end of the day, parents all have problems somehow and you can get rid of them or find solutions, yes (P5).*


*Exchange among young carers,* such as in the context of afternoon sibling meetings, is perceived as supportive. Such meetings may take place with or without the teacher’s support. Again, parents emphasize the importance of knowing who can provide professional support at school. Some parents suggest the potential relevance of *inclusive schools* in the context of sibling meetings, as all students encounter children with disabilities. Many of these students are affected siblings themselves, which makes exchange among them easier and more accessible. *Support services should be located at school*, they stress, due to the distance of extracurricular activities from students’ homes.

Furthermore, some parents highlight the need for school-based homework assistance programs. Thus, one father whose mother tongue is not German points out that he cannot help his son with homework due to the language barrier, creating an additional burden for both parties.

**Bullying and how it is handled:** Bullying is a recurring theme in the interviews with parents as well. They perceive a risk of bullying associated with open communication about illness and disability. For example, they may see a siblings’ disability as a potential cause for bullying. One mother also shared that her daughter’s silence at school regarding her mother’s mental health concerns came from a fear of being bullied. Several parents also expressed concern about the mistreatment of disabled children in schools. These parents recognize the need for effective *anti-bullying prevention and intervention strategies at school.* This, they believe, requires further training for teachers:


*But the teachers, I don’t want to demonize them all, but I think they are simply overburdened. They are poorly trained in such things. Because when I think about it, this bullying in particular could really be so well controlled with training. That the teachers can really react, can also react preventively. And not then stand there at the end and say, “For God’s sake, what happened now?” (P7).*


Inclusive education also emerges as a theme in the parents’ interviews. Addressed in the context of raising awareness/educating children, disability and illness, and combating bullying, it is tied into the broader theme of supporting their children. Parents perceive the stigma experienced by people, both at school and in public. Inclusion, they note, provides opportunities for all children to interact with disability and illness, which facilitates awareness: 


*I think it’s important to raise awareness of the issue. So to bring more attention to what it means to be disabled. I think that maybe it comes with all these inclusion classes now, that they even … Yes, many have never seen a disabled child. One is always quite shocked when one sees statistically how many disabled children are actually in wheelchairs. And when you walk through the city, you never see a child in a wheelchair. […] I always find it quite frightening. You ask yourself, where are all the children? Are they all locked away? (P2).*


Education about disability and illness is regarded as essential for appropriate interaction with ill or disabled individuals and following to prevent bullying. Teachers are expected to address these issues as part of their professional responsibilities, though they often lack the necessary skills. Two parents expressed a need for comprehensive teacher training on the topics of disability and illness, including associated care responsibilities of children. They feel that teachers often fail to recognize the reality of their family situations due to a lack of awareness.

**Communication between parents and school:** Communication between parents and teachers is a major topic in the interviews. Indeed, *regular communication* is described as essential to successful support. While parents see phone calls, mobile messaging, or written notes in a booklet and other rapid forms of communication as positive, they suggest that interaction with teachers typically only takes place during scheduled parent–teacher meetings. Parents express dissatisfaction with the exclusive emphasis on subject learning during these meetings.

Parents stress the importance of establishing *regular and early communication* with schools. Their children may consciously or unconsciously withhold information from them regarding their school experiences, parents note, such that they only become aware of certain issues after they have escalated significantly:


*[All the teachers at the conference said:] This can’t go on, and YC3, she’s hitting.” She became aggressive for a while. I didn’t notice any of that. Only through this one session, which (.) moved a lot (.). I would have liked the way a little bit different [P3 laughs]. But that was the first time that the problem really came to light, what kind of problems she had as a result. YC3 never let that show. On the contrary, she really always went easy on me and always looked out for me. Yes, she actually tried to protect me (.). Which of course isn’t right at her age (P3).*


At this point, the communication between school and parents often breaks down. In response, parents call for *professional communication management* skills for teachers and stress the need for *unbiased communication.* Many parents feel responsible for maintaining regular contact with teachers. This can be demanding, as schools are not always easily reachable by phone or accessible. It is important to *consider the possible limitations of family members* for successful communication, they stress. Parents do not want to constantly negotiate methods of communication.

Another requirement for successful communication, as mentioned by parents, is the *genuine interest of teachers*. During parent–teacher conversations, the focus is often on subject learning. However, parents also express a wish for teachers to ask about and allocate an *appropriate amount of time for communicating about the family situation*.

In comparison to their children’s current secondary school teachers, parents note, their primary school teachers *knew more about the family situation*. Home visits by primary school teachers were appreciated and interpreted as a show of interest in the student’s life and situation. Despite this, parents expressed understanding of the challenges that come with such visits, recognizing that teachers must divide their time among several students. In general, the parents acknowledge that the heavy workload of many teachers adversely affects the level of support they can provide to their students. 

#### 3.1.3. Perspective of the Teachers

**Classroom context:** The teachers interviewed describe different ways to support young carers in their classrooms. They fundamentally emphasize the importance of *recognizing every child as an individual and perceiving them through their strengths, regardless of their life situation*. They see in young carers *individual skills* that could be used in class:


*Things like mediation, students conflict mediator, class mentors maybe, that they might do something like that more (T4).*


Teachers also recognize that young carers have needs in the classroom beyond their caregiving roles. *They support young carers in the classroom to the extent that they are able,* for example by providing separate study areas for those who struggle to concentrate, or through personal help with the organization of studies.

Different perspectives on dealing with young carers emerge from the interviews. T4 emphasizes the importance of equal treatment:


*The child probably doesn’t want to do that either. And just this effect occurs, according to the motto: “You already have it so hard, it’s probably good if you only do half of the tasks.” And that can’t be it either. I also believe that the child, or the student, may no longer really feel valued, I would say.*


The other teachers share concerns about preferential treatment, despite acknowledging the persistence of such behavior in daily life:


*I don’t know if the mildness-bonus is so necessarily good. [T1-Reg, (regular) classroom teacher: No!] It’s not really, but you kind of do (T1-SNE, special needs teacher).*


On the one hand, special treatment for young carers manifests as leniency towards their behavior; on the other, it involves attentive support of their needs.

The teachers emphasize a *knowledge of the life circumstances* of young carers as a basis to be able to classify their behavior. They gain an insight into the life of the young carers through telling stories from the weekend. But these narratives decrease with increasing grade level in favor of subject learning. They recognize that the individuality and the awareness of social competences of one child disappear. Individual conversations and concrete enquiries with individual students are also becoming less frequent. T1-SNE and T1-Reg recognize this fact during the interview:


*Yes, we can ask that (…). No, that’s really the case. We could also ask YC1: What do you actually need? What would you like to have? What do you really enjoy? I haven’t asked that for a long time somehow (T1-SNE).*


A “simple” enquiry about the emotional state is perceived positively. However, teachers differ with respect to how much they ask about the family situation. T4 comments:


*Well, I’m not the kind of teacher who always approaches the students and interrogates. I don’t like that. And I think most of the students don’t like that either. I make offers and the students know that, I think, at least I hope so. But I wouldn’t go to YC4, for example, and say: “Listen, how are things at home?” However, you rather get to know that during the learning counseling day or yes, I hope that he might come by himself (…) The students then tell more than when you always ask. Then you would get mostly yes or no answers. [T4 laughs] Exactly.*


Teachers also stress the importance of setting an example of tolerance and of teaching tolerance for all family models and living situations. Project formats are often used by teachers to *address the topic* of young carers. T4 explains:


*It doesn’t have to be related to one’s own family. You might also talk about certain illnesses in science lessons, for example. That you get the consent of the student beforehand, of course, whether he or she might want to talk about something. Yes, and perhaps also in the sense of inclusion. Especially if the parents are physically impaired, this opens up the perspective of the other students a bit. Also, in dealing with sick people and people in need of care, I think that would also be very important. And that would perhaps also be a point where one could and should address this even more, yes.”*


**School context:** While teachers provide detailed insights about supporting young carers and in the classroom, many of their statements reflect broader, school-level context concerns.

They see an opportunity to *refer young carers to existing school structures* (such as homework supervision or school social work). *School social workers are seen as particularly helpful* due to their ability to form trusting relationships with young carers, unimpeded by the dynamics of grading.

Teachers acknowledge their responsibility for informing their teaching colleagues about special regulations or arrangements (such as educational accommodations) concerning young carers. The *exchange of information between teachers* is important, they note, but difficult to carry out in practice. In some schools, communication takes place automatically during the transition from primary to secondary school or when there is a change of classroom teacher. Without such communication, valuable knowledge and possible support for the young carers and their families may be lost. Home visits are described as invaluable for gaining deeper insight into children’s family environments.

A recurring theme in the interviews is the lack of *standardized guidelines or process plans for supporting young carers*. Much of the support happens on an individual level, which increases the teacher’s own workload.

**Communication between parents and school:** The theme of parental involvement in school and communication is also prominent in the interviews with teachers. All teachers emphasize the importance of regular communication for successful school attendance of young carers. Moreover, they highlight the preventive aspect of *early and regular communication*. *Individual communication channels and flexible contact times*, they stress, are more necessary for the parents of young carers than they are for other parents.

While teachers also see *awareness of the family situation* as a basis for support, they are aware that regular communication with parents is about academic performance. Some teachers see the difficulty in communicating about family matters in terms of how far they can intrude into families’ private lives. Even if they have positive intentions, there is a danger of denying parents skills:


*[…] These are two adult people, that is P4 and his wife, they are competent, and they live their lives and they are not, they can still do everything: raising their kids and take care of themselves and they are also very loving parents and I think you always have to be careful that you don’t deny them their life skills (T4).*


Teachers perceive that parents are often skeptical of the school as an institution, which limits open communication between the parties involved. One teacher associates limited communication from parents of young carers to concerns about dealing with them differently:


*However, I also think that parents are always afraid that they run risks that their child is put in such, such a light and that is also really difficult (T4).*


**External support systems:** All teachers recognize that schools are often *institutional first responders when it comes to supporting young carers*. Yet they also locate this support for young carers outside of school, for example in the form of an afternoon peer exchange. Finally, teachers see the *need for external support systems*, like community work or a youth welfare office, and stress that these should be present in schools.

#### 3.1.4. Perspective of the Counselors

**Classroom context:** Counselors first underscore the importance of *teachers recognizing the reality of young carers in their classroom*. From their perspective, the overall aim of support should be to increase the well-being of young carers. Counselors see opportunities for *teachers to adapt the school day and overall school life* to better support young carers. Specifically, they note that teachers can allow students to take time off in particularly stressful situations, or extend or change examination times and formats. 

Counselors perceive in young carers *specific resources* that should be recognized and valued by teachers in the classroom.


*Then the resource, for example, to put your own needs on hold. Deferred gratification rewards or whatever, (…) social thinking like that. So these are all just not skills that (…) score in school. For which you get good grades” (C5).*


They see the importance of *maintaining an awareness of, and educate all children about, caregiving and illness*. One way to do this is to integrate it into lessons as one topic of study.

**School context:** Counselors state that the responsibility for recognizing young carers lies with the teachers. Many teachers, they note, overlook a child’s individual needs due to the structure of the school system, focusing only on grades and academic performance. They warn that identifying and supporting young carers requires more *knowledge about the students and their families*. This knowledge is often lost during transitions to other schools, other classes, or when the classroom teacher changes.

Counselors recognize early warning signs in young carers’ behavior that show they are experiencing stress. At this point, intervention by teachers becomes critical. They suggest using the school counseling center as a support resource in such cases. Counselors also see prevention as very important. However, in daily practice, teachers often turn to counseling only when the children’s problems are already intense. The counselors see a lack of awareness among regular school teachers as well as a lack of *competence in recognizing problem situations and initiating support*. The expertise for these topics is held by teachers with counseling training, special education teachers, or social workers. In order to support young carers in school, there is a need for *specifically aware and trained professionals* who can stay abreast of what is going on with young carers and who know their needs and support options:


*[…] that one works towards it, that one is well informed about it, that there is someone at the school who is well informed and well networked, who also periodically points out to his colleagues in the conferences, in the team meetings again and again that, for example, certain symptoms such as absence from school, homework is not done and so on, tiredness, so all these symptoms can indicate it. Because teachers don’t automatically think about it or are attentive to it. […] So that there would be someone who would really have a professional knowledge. That there is just like these victim protection and child protection officers and prevention officers someone in the school. The person who repeatedly and prayerfully reminds colleagues, and those who have already forgotten, that this exists and that he or she can be contacted in cases of suspicion or […] in cases where children are suspected of being involved. So in the case of children, where one suspects that there is a special burden. So I think that would be good, and then these people would either have to be linked to us, because we are an organization that is already committed to networking (C6).*


Currently, support is organized on an individual level in schools. Teachers lack guidance, resulting in missing or arbitrary forms of support. Counselors stress the importance of having *general guidelines for young carers in school*. They also see school staff as responsible for customizing these guidelines to the needs of individual students on a case-by-case basis.

For example, one counselor highlights the need to create individual action plans with young carers. A professional, they note, can work with the young carer to develop a home emergency action plan:


*So, yes, there are such possibilities of a setting for children of caregivers with mental illness. That, for example, situations could be talked through: If such and such is, then I can do such and such. […] And in this setting, with these two to three people it is also clear: Emergency note, if Mom has a severe episode of her depression, then we know, you call Aunt Emilia, but you can also call the, what do I know, the Hannah, the social pedagogue of the school, exactly. And then we make an appointment. Then you come to school, then you don’t go to class, then you come to the appointment, something like that, for example” (C1).*


To ensure transparency, the counselors emphasize, such a plan is discussed with all the teachers and staff at the school as well as with the family. Group activities for young carers is seen as a further concrete support measure that can be offered at the school level. Here, the advantage lies in the peer contact and peer-to-peer communication such formats offer. To lead the group activities, *professional contact people* who are known to the young carers are also needed.

**Communication between parents and school:** The counselors emphasize the importance of communication between school, parents, and young carers. If the usual communication channels cannot be used due to a barrier (such as a hearing impairment), the school professionals have to *look for individual solutions*. In these families, successful communication with parents is significant. Counselors see that parents are quickly stigmatized and denied competencies.


*Of course, I think you have to be careful not to stigmatize the family again and say: “Oh yes, the child is completely overtired again, it’s the parents’ fault again.” I think you always have to be careful about that (C3).*


One of their tasks is to support teachers in their work with parents. In addition to a role as a facilitator in conversations, counselors can train teachers in communication with parents. Teachers often lack the *tools to address sensitive and private issues*:


*I think it’s good to open up possibilities like that, what should a class or what should the guidance counselor or the SNE-coordinator or whatever have on hand. So that when they know that a child has parents with mental illness, they are well prepared. So something like a checklist or something. That might not be bad. We haven’t developed anything like that, but I could imagine something like that that could perhaps be A a low-threshold input and B also something that could be included at the end or somewhere within a training course, i.e., a toolkit (C1).*


**External support systems:** If families of young carers are experiencing massive problems, there is a need for external support. In Germany, one difficulty is the wide range of institutional differences when it comes to support structures. Often, teachers do not know where to find helpful support. It is thus essential that schools develop a *list of available support structures* and that they integrate it into their *guidelines for supporting young carers*. Some counselors see it as their duty to support the school in this matter.

Counselors would like regional support systems to advertise their activities more actively. These *systems must actively contact each school*. For successful cooperation, every institution has to name a contact person. To facilitate communication, structures have to be transparent. 

### 3.2. Model of Basic Conditions of the Support Structures for Young Carers in School

To determine the basic conditions for successful support in school for young carers, the results presented above were analyzed and, in a second step, interpreted collectively across all perspectives. Subcategories were summarized and integrated in order to establish the core conditions for support in school, which we refer to as the *basic conditions.* These findings were collated across all perspectives and then visualized in a framework model. The Model of Basic Conditions of the Support Structures for Young Carers in School (Figure 1) and its components, consisting of key prerequisites and factors that enable success of support structures, are outlined below. This model represents the conditions deemed essential to potential school-based support mechanisms by one or more participant groups.

In accordance with Bronfenbrenner’s ecological theory of multiple levels [36], this analysis distinguishes between three main levels of support: the family level, the classroom level, and the school level. In line with our focus on schools and in order to structure the variation in our results, we categorize these factors as belonging to either the *school level* or the *class level.* The family level profoundly influences many of these factors, as examined in the corresponding sections.

At the school level, the *development of formal policies and procedures to support young carers* is a key aspiration of both teachers and counselors. Currently, teachers must devise solutions on a case-by-case basis. This takes extensive time and energy—time and energy that teachers often lack. To facilitate support, guidelines can serve as needed structures. The counselors also highlight that general guidelines for young carers in schools would offer a needed basis that individual schools can adapt to fit their own framework conditions and specific support measures. This allows schools to assess their available resources in order to use them effectively or to expand them if necessary. 

Developing formal policy and procedures can also serve as the basis for specific support structures, such as group activities for young carers. Parents consider these opportunities for exchange among other young carers as important for their children. They note that formal policies and procedures can also provide a structure for the organization of school-based information. In addition, counselors see the need for a list of support structures that is continuously available to school staff.

Closely linked to this is the factor of naming a contact person in the school environment. The need for a *designated contact person* to provide support is clear for all groups of participants, except the young carers themselves. Teachers identify school social workers as a possible professional group that could support young carers, viewing themselves as a more general point of contact for all students. However, they also recognize the need for trained school specialists for young carers. This need for specifically aware and trained professionals is seconded by counselors. The designated contact person is tasked with various support responsibilities, described by counselors as knowledge multipliers and networkers for external support who disseminate targeted information material and develop formal policies and procedures to support young carers. Such professionals must be known to everyone in the school environment so that they can be contacted if necessary. Parents, in particular, stress the need to ensure this person’s visibility and accessibility. While teachers and counselors discuss the need for support from specifically trained staff, young carers themselves tend to view their classroom teachers as their primary sources of support. Classroom teachers are on students’ “perceptive radar” as familiar figures whose role is known to them, unlike other types of professionals. While the need to raise a general awareness about the situation of young carers among teachers is clear, parents also stress the need for trained professionals who are present and known in schools and who are attuned to the specific needs of young carers.

Along with the task of networking via a designated contact person, *access to external support systems* emerges as another factor that enables the success of support structures. Parents, teachers, and counselors stress the importance of bringing existing external sources of support into the school. Teachers, in particular, emphasize the need for external professionals and extracurricular activities for young carers and the need for schools to strengthen cooperation with external support systems. Negative experiences and associations with the youth welfare office are evident in many of the interviews with professionals. However, a large number of regional support services are also mentioned. Since these are not always known, teachers and counselors highlight the importance of a communication dynamic in which external support systems actively contact the school. The designated contact person could serve as the person responsible for networking and for developing support structures in cooperation with these external support systems, which are needed by all stakeholders within the school environment.

*Action against bullying and discrimination* at a school level are dominant themes in the interviews with young carers and parents, but not those with teachers or counselors. Bullying was not a prepared topic in the interview guidelines, but the issue emerged during the course of the open-ended interviews with some of the young carers and their parents. Young carers and parents report negative experiences that affect themselves or family members. They also notice bullying of other students with disabilities or illnesses. Here, illness or disability is perceived as the reason for the bullying or as one aspect of a multi-causal structure. In addition, the young carers report the use of swear words and language with negative connotations in communications about disability. Even if they are not directly affected in person, the explicit desire for non-discriminatory language at school is evident. Another perspective on discrimination reported by parents is the child’s or the child’s relative’s lack of opportunities to participate in school life. They list the physical barriers that prevent these children from participating in social activities. Young carers also see this barrier if their siblings are affected. For this reason, it is important to recognize discriminatory situations for young carers, parents, and families. It also seems important that school staff identify the barriers and, in the best-case scenario, remove them.

In this context, parents emphasize the importance of inclusive schooling as a fundamental structure for countering bullying and discrimination. School inclusion is interpreted as offering students the opportunity to come into contact with the topic of disability itself. Parents and young carers state that teachers have a special role in inclusion. They lead by example, educating and intervening. Their current role in implementing school inclusion for students with disabilities is seen negatively. 

At the school level, one factor included in the model was only named by one group of participants, but it is relevant in conjunction with other factors that enable success of support structures. In the interviews with teachers, the importance of *teacher-to-teacher sharing of general and context-specific information* becomes clear. Teachers point out that this communication is sometimes not successful, for example during a student’s transitional phases in school. As a result, awareness of the family situation and established support is lost. The knowledge transfer factor appears to be more fundamental, as teachers report not knowing how to address sensitive issues. Some teachers report positive experiences of anchored handover talks that enable this knowledge transfer.

Further factors that enable success of support for young carers become clear at a classroom level. Firstly, all stakeholders mention factors that can be summarized under the rubric “*flexibility to develop tailored responses*” by teachers. Young carers and parents point out their wish to be treated with consideration. Parents, in particular, express concern about whether teachers will respond appropriately, stressing that teachers can only react if they have sufficient information about the student’s situation. Even informed teachers, parents report, fail to proactively initiate conversations with young carers and to react appropriately to their behavior in school. Response to behavior is closely related to the person’s attitudes towards special treatment. Parents reject the special treatment of their child but express the wish for teachers to change how they deal with the situation, if they are aware of it. Even if this can be interpreted as the same action from an outside perspective, for the people concerned, there would appear to be a difference. They, as well as the counselors, see the need to train regular school teachers on how to handle challenging behavior. All stakeholder groups report situations where teachers are currently already providing support in the classroom. In particular, young carers and parents are aware of classroom-level support. Teachers are seen as having options to adapt the school day and school life to fit the needs of young carers (for example, by reducing homework).

Some teachers point out that they have limited resources to adequately handle every individual case due to their heavy workload. This factor is also acknowledged by parents and counselors. As mentioned previously, developing formal policies and procedures to support young carers could help to establish the kinds of procedures and processes that can guide teachers in their efforts to support young carers. For example, guidelines could be used to identify actionable areas, regulate information sharing, or provide guidance on how to involve other people or systems in a specific case.

In connection with this factor of the school level, the *awareness of policy and how to access support* by all teachers is a necessary structure. Teachers and counselors emphasize the importance of this awareness to identify and support young carers in the classroom to the extent possible. The better they know the framework conditions and possibilities, the better they can adapt and utilize them individually in the classroom. According to counselors, teachers also need skills that will enable them to recognize problem situations and initiate support.

Parents, teachers, and counselors see particular abilities and competencies of young carers, which they relate variously to the varying situations of these children. They point out that these skills are not directly relevant to grades but can and should be valued in different ways at school. Hence, *positively recognizing young carers’ attributes and competencies* is highly rated as vital to establishing a supportive school environment. Talks at school are mentioned in the interviews of some stakeholder groups in connection with perceived strengths. Teachers see giving presentations as a strength among young carers, one that can be used in the classroom to help educate their peers. While teachers recognize this as an opportunity to harness young carers’ knowledge and experience, they emphasize that this requires individual consultation. The latter is viewed as highly relevant, as some parents report that teachers asked their child, as supposed experts, to talk about family issues in front of their peers without having been asked beforehand.

Peers play an important role in the school life of young carers. Often, their role is addressed in the interviews in the context of rejection and bullying. For all stakeholders, *raising peer group awareness of issues and challenges for young carers* is perceived as an important basis for behavioral change. In addition to peer education, however, school must also be the place where young carers receive information and education. This focus is particularly evident among parents who recognize their children’s difficulties and fears at school, which they do not address at home. The need for teachers to address the topic in school by raising awareness and educating all children about caregiving and illness is, therefore, a relevant factor that enables the success of support structures for young carers in school. 

Another factor that comes up in the interviews with parents and both professional groups is effective communication between parents and teachers. At present, such communication is generally deemed inadequate and unsuccessful.

Parents express a desire for regular and early communication, as well as a wish that teachers will take a genuine interest in their children. An appropriate framework for discussing the school and the family situation with teachers is seen by parents as crucial. Early communication, in particular, can prevent or avert problems at school. This assessment is made based on the negative experiences of parents, who stress the need for teachers to be trained in professional communication, including *unbiased communication*.

Teachers also see successful communication with parents as the basis for supporting young carers. Like the parents, they rate early and regular communication as important. However, they are aware that conversations are often about school performance and not the family situation. Teachers express their difficulty in communicating private matters with families. Parents and counselors are also aware of this. The teachers assume that parents do not communicate everything for fear that their child will be treated differently.

Counselors also see the need for teachers to receive further training in communication skills as a condition for successful communication. Particular attention is paid to conversational skills to address sensitive and private issues. They also formulate the possibility that counselors or school social workers can take part in discussions.

For all of the factors examined, the need for an *awareness of individual experiences and family backgrounds* for adequate support at school stands out as a key prerequisite. This includes not only an awareness of the young carer as part of the family, but also of the family’s situation as a whole. Awareness of the caregiving students, as well as their family situations and life circumstances, form the primary base of school support structures for young carers. For schools, recognizing the reality of young carers requires a general *awareness of issues affecting young carers*. Although parents, counselors, and teachers believe that teachers must be aware of and support all students, they also point to the need to recognize special characteristics of young carers, as well as warning signs and changes in their behavior. This, in turn, presupposes that teachers see pupils as individuals. Parents, in particular, stress that tailored support for this target group can only take place through awareness.

Awareness of the issues affecting young carers and the individual experiences and family background are seen to form a framework that both facilitates school support for young carers and influences the quality of those support structures. However, it becomes clear that support at the school or class levels can also increase the level of awareness regarding the phenomenon itself, and awareness of the family situation. 

When analyzing individual factors that enable success of support structures, these cannot be viewed in isolation; rather, they influence each other. This also applies to the classification of conditions as either “class level” or “school level”, as there are always potential overlaps.

## 4. Discussion

The model shows the basic conditions for support structures for young carers in schools, integrating the perspectives of young carers, parents, teachers, and counselors.

Raising awareness on the issues affecting young carers emerges as a key factor in the model. Such awareness clearly stands out as the basis for identifying young carers in the first place. The model also shows the importance of seeing and acknowledging young carers’ individual experiences and family backgrounds and stresses the crucial importance of this understanding for acknowledging young carers in schools as individuals who belong to larger family constellations. In this model, all factors that enable success of support are influenced by awareness, but they also influence awareness itself.

Raising professionals’ awareness increases the visibility of young carers, which is crucial for their health-related quality of life and mental health [38]. Despite their vital importance, programs and resources dedicated to educating professionals working in the fields of education, health, and social services for young carers are currently inadequate. Levels of awareness and visibility of young carers vary internationally [5,17]. In Germany, while the significance of the phenomenon remains high, societal awareness and specific support are not widespread [5]. The counselors in this study question how teachers can be expected to be aware of young carers if the broader society lacks such awareness. Other experts debate the extent to which schools should be responsible for addressing young carers’ needs and issues [17]. It is increasingly evident that raising awareness is essential at all levels of society, especially among professionals in education, health, and social care.

For effective support of young carers, a comprehensive understanding and awareness of every young carer’s unique situation is essential. Designating a specific contact person and formal policy and procedures in school are also of key relevance. This model identifies factors that can serve as starting points for support: adopting measures against bullying and discrimination, fostering teacher-to-teacher information sharing, and maintaining the flexibility to develop tailored responses for individual students in the classroom are more general forms of support that can be helpful for all students. In particular, these measures can be crucial for providing support to young carers who remain “hidden from view” [39].

In Germany, this model is considered novel and unique due to its support of young carers in school. In contrast to other countries, which already have guidelines or designated contacts at the school level, this is not yet the case in Germany. It is clear that recognizing and acknowledging young carers as a social phenomenon is critically important. Since support can begin at the classroom level, it is crucial that classroom teachers have the skills to identify and initiate support for young carers. Communication between teachers and other school staff is key at this level. Classroom teachers tend to hold significant information about individual families and share a special bond of trust with them. Therefore, they play a vital role in linking young carers to other professionals. Given their significance in the lives of young carers, it is imperative that teachers receive training and further education that will enable them to fulfill this responsibility [21]. The participants in our study suggest the need for such training, in accordance with Nap et al. [17]. 

The need for further education and training is evident in another critical area: communication between parents and teachers. All stakeholders mention communication at various levels. In so doing, they refer to the type and method of communication within and between different stakeholder groups, alongside time and space considerations. The content of the communication is also discussed. In particular, communication between parents and teachers is often discussed in terms of helplessness on both sides. Parents show a willingness to engage with their child’s classroom teachers about topics relevant to their family and acknowledge their responsibility to keep an open dialogue with teachers. Nevertheless, they still maintain their privacy and do not disclose everything. Teachers and counselors are aware of this fact but do not bring it up in their interaction with parents. This can lead to a situation where teachers and counselors assume that parents are unable or unwilling to communicate everything, even when this is not the case. However, parents, teachers, and counselors recognize the need to address specific topics within the family in order to understand behavior and provide appropriate support. The extent to which teachers may enter the private sphere is not established, but subjective perceptions have led to unwarranted assumptions in this respect. As teachers are often in the role of initiating conversations, it is essential that they possess technical know-how and abilities concerning dialogue techniques, along with subject-specific know-how regarding young carers [21]. Other professionals or specialists, such as the named contact person for issues concerning young carers, can support teachers during the conversations or help them prepare for such encounters [21].

Our study finds that individuals with visible disabilities or illnesses are at a higher risk of being bullied, supporting previous international research [40,41]. It is imperative that schools implement and extend anti-bullying techniques and policies, while also increasing awareness among peers of the challenges faced by young carers. Teachers bear a significant responsibility in tackling bullying and discrimination and must improve their handling of these issues. Notably, school inclusion has been suggested by some parents and young carers as a means of addressing such problems. Although the reality of school inclusion is currently perceived differently, some parents believe that peer-to-peer contact at school among students with illness or disabilities leads to more positive attitudes and decreases prejudice towards these students. They also see the importance of providing information about illness and disability in order to reduce fear of contact, combat prejudices, and explain behavioral issues. One way that young carers can serve as role models is by giving talks and presentations to their peers. This form of participation is a way to acknowledge and appreciate the unique skills of young carers. Nonetheless, such an initiative must be discussed with students, parents, and teachers on a case-by-case basis to avoid overburdening or exposing these children to undue stress. Parents’ assumption that contact and information can be fostered in peers’ positive attitudes towards students with disabilities is backed by research [42,43]. This link is crucial not only for the support of young carers at school, but also for the promotion of inclusive attitudes. While parents recognize that perfection is unattainable, they underscore the importance of cultivating attitudes of acceptance towards people with disabilities and of identifying appropriate solutions where necessary—an effort that primarily involves communication. They acknowledge that not all issues can be resolved in reality. Nevertheless, parents request that teachers and schools actively communicate whenever an obstacle or problem arises.

If schools are to provide adequate support to young carers and their families, they must broaden their horizons to include other institutions. Integrating support systems into school life is crucial and requires mutual willingness, intention, and practical execution. Not only would this initiative facilitate access to support for all stakeholders; it would also lay the groundwork for enhancing relationships between different professions, which can often be challenging [44]. This approach is exemplified by the whole-family paradigm [25] and the Young Carers Support Model [23], which view cooperation between all involved parties as essential. Here, a holistic approach engaging the family and professionals from the education, health, and social service sectors is deemed necessary for facilitating the sharing and dissemination of knowledge [17]. Importantly, it offers not only an intervention for problematic situations, but also preventative measures for families in care settings. Employing the guidelines outlined in the model can enhance the collaborative effect while enabling an individualized and systematic approach.

When evaluating the model resulting from this study, several contextual factors need to be considered. These include the purposive selection of participants, the sample size, and the inherent subjectivity of their responses. Importantly, all of the respondents are aware of young carers; that is, none of the interviewed young carers were invisible in their situations. Our secondary analysis of 29 interviews is grounded in two qualitative studies that focus on young carers and schools, which differ in emphasis. Research Project 2 focuses primarily on external counseling resources within the educational context, rather than everyday school life. In the present study, our analysis of counselors’ perspectives offers valuable insights into the school lives and school-based support systems of young carers. The model was developed using qualitative data, which combined the subjective perspectives of young carers, parents, teachers, and counselors. The low level of awareness and frequent absence of support for young carers in German schools should be considered when contextualizing the results. Additionally, it is crucial to consider the influence of peers and other professionals, such as social workers, within the school environment and creation of support. The model has not considered these factors. These limitations should be kept in mind when evaluating and categorizing the model.

## 5. Conclusions

As an institution, school plays a significant role in supporting young carers. School problems are diverse, and situations are unique to each individual, although there are some similarities among young carers. For support mechanisms to be successful in this setting, a number of conditions are necessary. Studies indicate that specific programs are not always needed [17]; instead, integrating support for young carers into existing structures is crucial to ensure the sustainability of the assistance over time [17]. The Model of the Basic Conditions of Support Structures for Young Carers in School presented and discussed here provides initial insights into basic approaches that can be adapted to support young carers at school. While some conditions are specific to young carers, others are more general and can preventatively impact various student groups at school. These conditions also apply to young carers who remain “invisible” to professionals. Within this model, students who are young carers are perceived individually, yet their situations are not considered in isolation. Rather, they are seen as part of a broader system in which parents and teachers also play significant roles.

The model shows that teachers bear a significant responsibility for providing effective support at school. As the first point of contact for students, and as professionals who know their families, teachers can act as gatekeepers for further support. Nevertheless, they must not be left alone with this responsibility; support from other school-based professions is vital. In addition to providing teachers with suitable training, ongoing education programs, and adequate time resources, it is essential to facilitate connections between teachers and external support resources. As these resources must be available and present, networking with external support professionals is crucial. It is also imperative for schools to establish specific guidelines regarding these resources to ensure that all available support options are transparently known to and can be effectively utilized by school professionals. This fact is stressed by school counselors, whose viewpoint has not yet been considered in research. Despite their vital support function in schools, counselors acknowledge that they are seldom contacted by teachers—typically only in cases when a student’s issues are significant. These findings suggest that improving basic conditions by building on existing (and generally non-specific) structures can help to reduce the workload for teachers and provide targeted support for young carers.

In light of these findings, there is a deep need to increase societal awareness of the issues faced by young carers and their families, as well as to establish societal support structures for these individuals. This is especially true in Germany, where this study’s data originated. As in other countries, the challenges faced by young carers in German schools are evident [18], but the level of in-country awareness and the policy responses to young carers can be described as preliminary [5,24]. Further research is required. For this, it is crucial that Germany works at a structural level when developing potential new policies.

When developing the present model for support systems within the educational sector, Nap et al.’s four criteria [17] for identifying the failure of support structures in the welfare domain were considered through the process. These four criteria are as follows: “(1) Interventions not matching the needs of (A)YC’s, (2) good interventions that remain underused because people are not familiar with them, (3) a lack of research to substantiate the effectiveness of interventions in the welfare domain, and (4) lack of capacity or finances to arrange formal support programmes” [17] (p. 10). In particular, criteria (2) and (4) show that existing structures in schools can be utilized and expanded without creating parallel structures. Professionals are needed to tackle the issue and provide a concrete list of the options already available for each school. Results were analyzed and discussed using international studies. However, it is crucial to assess whether the support structures and interventions align with the actual needs of young carers (1). The effectiveness (3) of the model must also be evaluated.

This secondary analysis first examined and described the differing perceptions of school support among young carers, parents, teachers, and counselors. These perspectives were then integrated into a single model in order to gain a deep, multi-perspective view of the topic. 

In so doing, this study gave voice to key actors in the school environment whose perspectives have been underrepresented in research. As a further step, it would be necessary to survey the peers of young carers, as well as other staff groups (such as school social workers) regarding their views on young carers and school support. Integrating these new perspectives into the model could help schools expand their support resources for young carers and may enable researchers uncover other obstacles to identifying young carers and designing effective support mechanisms.

Overall, this integrated model provides a foundational framework for supporting young carers in the school environment. Schools offer many starting points for initiating low-threshold and preventative actions to preclude school-related problems. By building on these existing and generally non-specific structures, the model offers the opportunity to create sustainable, school-based support structures for young carers that are also useful for other target groups.

## Figures and Tables

**Figure 1 healthcare-12-01143-f001:**
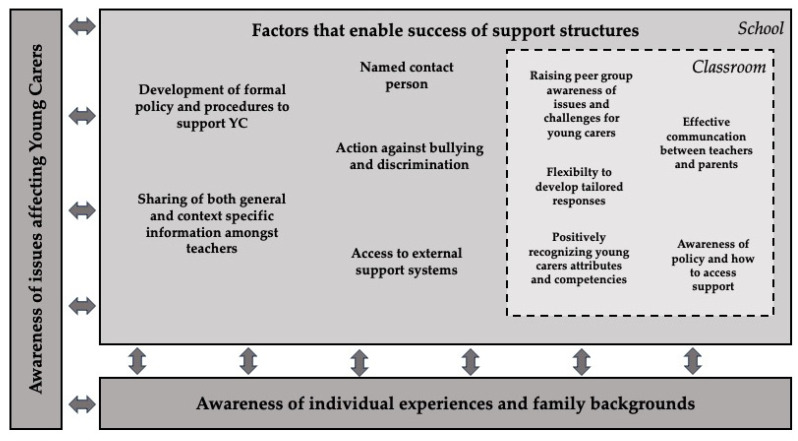
Model of Basic Conditions of the Support Structures for Young Carers in School.

**Table 1 healthcare-12-01143-t001:** Main categories.

Young Carers (YC)	Parents (P)	Teachers (T)	Counselors (C)
(1) classroom context	(1) classroom context	(1) classroom context	(1) classroom context
	(2) school context	(2) school context	(2) school context
(3) bullying and how it is handled	(3) bullying and how it is handled		
	(4) communication between parents and school	(4) communication between parents and school	(4) communication between parents and school
		(5) external support systems	(5) external support systems

## Data Availability

Data from this study are available on request from the corresponding author. The data are not publicly available due to privacy or ethical restrictions.

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
