# Peer review of "Basic Conditions for Support of Young Carers in School: A Secondary Analysis of the Perspectives of Young Carers, Parents, Teachers, and Counselors"

_healthcare, 2024, doi:10.3390/healthcare12111143_

Round 1

Reviewer 1 Report

Comments and Suggestions for Authors

Dear authors,

What an interesting secondary analysis you have conducted! I thoroughly enjoyed reading your study and it's findings. I have a few suggestions that I think could strengthen the article.

Methods - could you add if the interviews were conducted in German or English and if there was any translation (how do you translate the quotes for instance, for an English audience).

Results:

One, I encourage you to reconsider the use of the word "issues" - such as "classroom issues" and "school issues". In most English speaking countries, these are seen only as negative, yet many positive examples are highlighted. Maybe using the terms "classroom context" and "school context" more adequately describes how you are organizing these findings?

Two, I feel there are three levels that need to be described. If one considers how Bronfenbrenner structured his model, the school and classroom levels are second and third to the family level. Here you could offer solutions for families to have ways to communicate with teachers about their needs for their young carer within the school context, and findings such as a needs for homework support. The teachers mentioned home visits - this could be included in the family level as a way teachers can become more educated about the family's needs in a setting that is not academic. 

Line 344-346 seems like it belongs at the "school level" rather than about bullying. 

Overall, this is an excellent review. The reason I bring up identifying the family-level items and to clarify a few other things, such as perhaps home visits are common in Germany, are to help readers from other countries translate this into their contexts. The world is becoming so global, so identifying homework help for caregiving children of non-original language speakers for one, is key, along with the policy and awareness you describe so beautifully. Just a few tweaks here and there and readers from other countries can take this article to their governments and school districts and advocate for similar supports. 

Reviewer 2 Report

Comments and Suggestions for Authors

Lines 29-33: I suggest that you avoid using bullet points here- please revise this section

Can you provide a definition of ‘young carers’ somewhere in the introduction?

Your rationale in the introduction should explicitly state the lack of evidence from studies specifically in Germany. Can you please add this in somewhere?

Lines 104-105: Were the interviews conducted online or in person? Provide some details about where the interviews were conducted. Were the interviews recorded? This may be a secondary analysis but inclusion of these details is key to the methodology. Were the interviews conducted in German or in English? Were the transcribed verbatim? How was this done and by whom?

Line 129: Same comment as the one above for research project 2. Can you please include these details about the interviews?

Line 162: Did you provide an incentive for participation in the study?

Line 162: Provide the ethics approval number (or equivalent).

Given the already high load of teachers, how pragmatic is it for teachers to also support young carers? What kind of support can the teachers be expected to provide?

Reviewer 3 Report

Comments and Suggestions for Authors

The authors bring a very interesting topic that is good to publish.

However, it would be best to modify the article.

The introduction to the topic is missing:

- characteristics of young caregivers (nowhere is this term explained, supported by literature, previous research, etc.)

- research questions are also missing

- research - this is secondary data processing. It's alright. However, the selection of two groups of respondents needs to be clarified. At first, these groups can be significantly different (the researches are relatively far apart in time). Have you considered, for example, the possible development of the issue? It would be good to clarify this fact.

Round 2

Reviewer 3 Report

Comments and Suggestions for Authors

The authors have made recommended edits.